# Cultures of Human Skin Mast Cells, an Attractive In Vitro Model for Studies of Human Mast Cell Biology

**DOI:** 10.3390/cells13010098

**Published:** 2024-01-02

**Authors:** Srinivas Akula, Shiva Raj Tripathi, Kristin Franke, Sara Wernersson, Magda Babina, Lars Hellman

**Affiliations:** 1Department of Cell and Molecular Biology, Uppsala University, The Biomedical Center, Box 596, SE-75124 Uppsala, Sweden; srinivas.akula@icm.uu.se; 2Department of Anatomy, Physiology, and Biochemistry, Swedish University of Agricultural Sciences, Box 7011, SE-75007 Uppsala, Sweden; sara.wernersson@slu.se; 3Institute of Allergology, Charité—Universitätsmedizin Berlin, Corporate Member of Freie Universität Berlin and Humboldt-Universität zu Berlin, Hindenburgdamm 30, 12203 Berlin, Germany; shiva-raj.tripathi@charite.de (S.R.T.); kristin.franke@charite.de (K.F.); 4Fraunhofer Institute for Translational Medicine and Pharmacology ITMP, Immunology and Allergology IA, Hindenburgdamm 30, 12203 Berlin, Germany

**Keywords:** mast cells, granule proteases, cathepsin G, tryptase, chymase, Fc receptors, IgE, prostaglandins, leukotrienes, heparin

## Abstract

Studies of mast cell biology are dependent on relevant and validated in vitro models. Here, we present detailed information concerning the phenotype of both freshly isolated human skin mast cells (MCs) and of in vitro cultures of these cells that were obtained by analyzing their total transcriptome. Transcript levels of MC-related granule proteins and transcription factors were found to be remarkably stable over a 3-week culture period. Relatively modest changes were also seen for important cell surface receptors including the high-affinity receptor for IgE, FCER1A, the low-affinity receptor for IgG, FCGR2A, and the receptor for stem cell factor, KIT. FCGR2A was the only Fc receptor for IgG expressed by these cells. The IgE receptor increased by 2–5-fold and an approximately 10-fold reduction in the expression of FCGR2A was observed most likely due to the cytokines, SCF and IL-4, used for expanding the cells. Comparisons of the present transcriptome against previously reported transcriptomes of mouse peritoneal MCs and mouse bone marrow-derived MCs (BMMCs) revealed both similarities and major differences. Strikingly, cathepsin G was the most highly expressed granule protease in human skin MCs, in contrast to the almost total absence of this protease in both mouse MCs. Transcript levels for the majority of cell surface receptors were also very low compared to the granule proteases in both mouse and human MCs, with a difference of almost two orders of magnitude. An almost total absence of T-cell granzymes was observed in human skin MCs, indicating that granzymes have no or only a minor role in human MC biology. Ex vivo skin MCs expressed high levels of selective immediate early genes and transcripts of heat shock proteins. In validation experiments, we determined that this expression was an inherent property of the cells and not the result of the isolation process. Three to four weeks in culture results in an induction of cell growth-related genes accompanying their expansion by 6–10-fold, which increases the number of cells for in vitro experiments. Collectively, we show that cultured human skin MCs resemble their ex vivo equivalents in many respects and are a more relevant in vitro model compared to mouse BMMCs for studies of MC biology, in particular human MC biology.

## 1. Introduction

Mast cells (MCs) are hematopoietic cells that are positioned at the interphase between tissue and the environment, such as the skin, the tongue, the intestinal mucosa, and the lung [1]. They are highly specialized cells filled with electron-dense granules that can act on external threats to initiate an inflammatory response. A number of physiologically highly potent substances, such as histamine and heparin, and a number of abundant granule proteases are stored in these granules. The granule-stored proteases can constitute up to 35% of the total cellular protein and the transcript levels for these proteases are among the highest in the entire cell [1,2]. The levels can reach several % of the total transcriptome [1,3]. Heparin and the related chondroitin sulfate are highly sulfated and negatively charged proteoglycans expressed by MCs. They play important roles as anticoagulants, as stabilizing agents for MC proteases, and as charge-compensating molecules in the granule storage of positively charged mediators, including histamine. The enzymes involved in the synthesis of these proteoglycans and of histamine are therefore important markers for MCs.

MCs can rapidly release their granule material upon stimulation of the cell by the crosslinking of receptors for immunoglobulin E (IgE) or by stimulation by complement components like the anaphylatoxins C3a, C4a, and C5a. MCs can also be activated by positively charged low-molecular-weight compounds, like substance P and compound 48/80. This response is mediated by the MRGPRX2 receptor, which is almost exclusively expressed by connective tissue-type MCs (CTMCs) [4,5,6]. Upon activation, MCs also produce a number of lipid mediators, including leukotriene C4 (LTC4) and prostaglandin D2 (PGD2). The enzymes involved in their synthesis, such as phospholipase A2 (PLA2), the hematopoietic prostaglandin D synthase (HPGDS), and leukotriene C4 synthase (LTC4S), are also important markers for MCs.

MC development and migration are regulated by cytokines and chemokines that deliver their information by binding to specific cell surface receptors. The growth and differentiation of MCs are primarily regulated by stem cell factor (SCF), interleukin-3 (IL-3) (especially in the mouse), IL-4, and IL-33, and the receptors for these cytokines are important characteristics of MCs [7,8,9,10,11,12,13,14,15,16,17,18,19,20,21,22].

Transcription factors play a key role in cell growth and differentiation and several of them are important characteristics of MCs, including GATA1, GATA2, and MITF [1,5,23,24,25]. Analyzing the transcript levels for these transcription factors can therefore reveal important features of MCs and how their phenotype changes upon in vitro culture conditions. These transcription factors are also important for the physiological role of MCs and can thereby give information on differences in MC biology between species. The MC-related transcription factors were therefore used in the present study, together with a panel of other proteins, as markers for the comparison between MCs in mouse and man and for the characterization of how MCs change upon culture. A selection of results from published studies on mouse peritoneal MCs and in vitro expanded mouse bone marrow-derived MCs (BMMCs), the two most commonly used models in mouse MC research, using the Ampliseq technology are summarized in Table 1 [1,26] for comparison with the human MC data presented in this article.

The majority of tissue MCs seem to originate from an early wave of cells coming from the yolk sac [27,28]. However, during inflammatory conditions, MC precursors can also be recruited from the bone marrow to increase in number in the inflamed tissue [29]. The fate of these cells upon returning to steady state conditions of the tissue is not fully known but most likely involves apoptosis of the majority of the cells. This relatively recent information concerning the origin of the majority of tissue MCs and macrophages has changed the view of the role of the bone marrow for both MCs and other hematopoietic cells like macrophages. Such information is of major importance for our view of the characteristics of in vitro-developed cells originating from the bone marrow compared to tissue-resident MCs and macrophages [30,31,32,33,34,35,36]. The most commonly used in vitro model for mouse MCs is namely bone marrow-derived MCs (BMMCs). These cells most likely represent cells of a different origin compared to the majority of tissue MCs, which appear to be yolk sac-derived. We have also recently shown that these BMMCs represent relatively immature cells that differ markedly in phenotype from tissue-resident MCs and should therefore be used with care as in vitro models of MC physiology [26] (see also Table 1). An interesting in vitro model for studies of MC physiology is therefore human skin MCs that can be expanded in vitro by a medium containing serum and recombinant SCF and IL-4, as described in this article. We can show here that they resemble tissue-resident MCs much better than mouse BMMCs and are therefore superior as an in vitro model, in particular for research of human MC biology.

## 2. Materials and Methods

### 2.1. Purification of Human Skin MCs

MC purification was performed as previously described [37,38] with several modifications specified in more recent work [21]. Skin was cut into strips and treated with dispase (24.5 mL per preparation, activity: 50 U/mL; Corning, Kaiserslautern, Germany) at 4 °C overnight. After removal of the epidermis, the dermis was chopped into small pieces and digested with 2.29 mg/mL collagenase (Worthington, Lakewood, NJ, USA), 0.75 mg/mL hyaluronidase (Sigma, Deisenhofen, Germany), DNase I at 10 µg/mL (both from Roche, Basel, Switzerland), and 5 mM MgSO_4_ for 1 h at 37 °C.

The cell suspensions were separated from remaining tissue by three steps of filtration. In the case of breast skin, the undigested tissue still remaining after the first digestion was subjected to a second digestion step of 1 h at 37 °C after the first filtration. MC purification from the dispersates was achieved by positive selection with anti-human c-Kit microbeads and an Auto-MACS separation device (both from Miltenyi Biotec, Bergisch Gladbach, Germany). MC purity always exceeded 98%, as assessed by acidic toluidine-blue staining (0.1% in 0.5 N HCl). Viability by trypan blue exclusion exceeded 99%. We used between 4.8 and 6.2 × 10^6^ MCs for one RNA isolation (ex vivo samples). Cultures were started from 1.5–6 × 10^6^ MCs; around 3 × 10^6^ were eventually used for one RNA preparation.

### 2.2. In Vitro Culture of Human Skin MCs

The isolated skin MCs were expanded in vitro for 2–3 weeks in cell culture medium (basal Iscove’s medium; BioSell, Feucht, Germany) containing 10% fetal calf serum (Biochrom, Berlin, Germany), 1% P/S (Fisher Scientific, Berlin, Germany), 1% non-essential amino acid solution (Roth, Karlsruhe, Germany), 100 ng/mL recombinant human stem cell factor (SCF) (Peprotech, Rocky Hill, NJ, USA), and 20 ng/mL of recombinant human IL-4 (Peprotech, Rocky Hill, NJ, USA). The medium was changed every three days to ensure there was a sufficient quantity of the recombinant cytokines.

### 2.3. RNA Isolation and Heparinase Treatment

Total RNA was prepared from freshly isolated MCs and in vitro-cultured MCs, following an established protocol for each preparation. Briefly, MCs were lysed in 700 µL QIAzol^®^lysis reagent (Qiagen, Hilden, Germany), mixed with 140 µL chloroform (Sigma) and 60 µL DEPC-treated water, and transferred to a 2 mL gel tube (Quanta bio/VWR, Dresden, Germany). After centrifugation, the supernatant was transferred to a NucleoSpin^®^ filter and RNA was isolated using the NucleoSpin RNA kit from Machery-Nagel (Düren, Germany) following the manufacturer’s instructions. For heparinase (BioLab, Braunschweig, Germany) treatment, the resulting RNA solution was mixed with RNAse inhibitor (Thermo Fisher Scientific) and heparinase buffer (BioLab, Braunschweig, Germany) and incubated for 3 h at 25 °C. Another RNA isolation procedure was followed, using the NucleoSpin RNA kit from Machery-Nagel (Düren, Germany) according to the manufacturer’s protocol. To further concentrate preparations, RNA was precipitated overnight at −80 °C using 100% ethanol and sodium acetate (Merck, Darmstadt, Germany). The RNA of each preparation was eventually solved in 20 µL DEPC-treated water. After each treatment step, RNA concentration was determined by using a Nanodrop ND-1000 (Nano Drop Technologies, Wilmington, DE, USA).

### 2.4. Ampliseq Analysis of the Total Transcriptome

The transcriptome of freshly isolated mast cells and the different cultures were analyzed for their total transcriptome by the Thermo-Fisher chip-based Ampliseq transcriptomic platform at the SciLife lab in Uppsala, Sweden (Ion-Torrent next-generation sequencing system—Thermofisher, Waltham, MA, USA). The sequence results were delivered in the form of Excel files with normalized expression levels for an easy comparison between samples. In the Ampliseq analysis, all transcripts are read only once; this is why no normalization is needed.

### 2.5. Validations and RT-qPCR

For the first approach, cultured skin MCs were stimulated with an anti-FcεRI-Ab, AER-37 (0.5 µg/mL) (Abcam, Cambridge, UK), for 30 min, 90 min, and 24 h and compared to cultured unstimulated skin MCs. In the second approach, skin MCs were harvested 1 and 2 d after isolation, as well as after 3 weeks of cultivation. For the final part, cultured skin MCs were exposed to enzymes identically to what is used for skin MC isolation (2.29 mg/mL collagenase, 0.75 mg/mL hyaluronidase, DNase I at 10 µg/mL, and 5 mM MgSO_4_) and kept at 37 °C with or without shaking prior to harvest. RNA was extracted with the NucleoSpin RNA kit (Macherey-Nagel, Düren, Germany) and reverse transcribed with TaqMan Reverse Transcription reagent (Applied Biosystems, Foster City, CA, USA) according to the manufacturer’s recommendations. Quantitative PCR was performed with the LightCycler^®^ FastStart DNA Master SYBR Green I reaction mix (Roche Diagnostics, Mannheim, Germany) in a QIAquant 96 5plex real-time PCR cycler (Qiagen, Hilden, Germany). The primers for target genes were designed with NCBI primer blast software and synthesized by TIB Molbiol, Berlin, Germany. The 2^−ΔΔCT^ method was used to quantify the relative expression levels of target genes to several reference genes (appearing at the end of the table, i.e., β-actin, GAPDH, HPRT, and Cyclophilin B = Peptidylprolyl Isomerase B, gene name PPIB), as described (Table 2) [39,40]. Statistical analysis and visualization were performed with GraphPad Prism software (La Jolla, CA, USA, version 10.0.3.)

## 3. Results

### 3.1. Samples Used for Analysis

Samples of human foreskin and breast skin were, after surgery, collected and digested with dispase and collagenase to obtain a single cell suspension. These cells were then subjected to purification by magnetic cell sorting using a non-activating anti-kit antibody. Approximately 98%-pure MCs were obtained by this procedure for analysis of their entire transcriptome, before and after in vitro culturing for 2–3 weeks (Figure 1). The in vitro culture was performed in a serum containing a medium to which recombinant human stem cell factor (SCF) and interleukin-4 (IL-4) had been added. The aim of cultures was to obtain more cells for in vitro studies of MC biology. The number of MCs increased about 6–10-fold, by a three-week culture [38].

For the freshly isolated MCs, we analyzed three independent samples, all isolated from foreskin. Due to the relatively few cells obtained from each foreskin, three separate donor pools encompassing material from 12, 7, and 6 individuals, respectively, were used for the three fresh samples. The cultured samples were from 5–15 foreskin donors (or individual donors in the case of breast skin). For the in vitro-expanded cells, we had four cultures in total, two originating from foreskin (cultured for 20 and 17 days) and two from breast skin, cultured for 18 and 19 days. These seven samples were analyzed by the Ampliseq technology to obtain a quantitative map of the expression levels of all approximately 21,000 human genes.

By analyzing the expression of one non-coding RNA involved in X chromosome inactivation, XIST, we confirmed the origin of the seven samples in this analysis. Two of the three freshly isolated samples and the two originating from cultured foreskin MCs were clearly negative for this transcript, showing that they were of male origin. The two samples from breast skin showed high levels of expression of this gene, confirming their female origin (Table 3). Interestingly, one of the freshly isolated samples had a low level of this transcript indicating a mixed phenotype (Table 3). However, there are also reports of XIST expression in male cells [41]. We also included a male-specific gene, RPS4Y1, which is expressed from the Y chromosome and therefore cannot be expressed by female cells. This gene was expressed at relatively low levels in all three freshly isolated samples and in the two foreskin samples but not in the two breast samples, confirming the male origin of the five foreskin samples (Table 3).

### 3.2. Transcript Levels for the Major Granule-Stored Proteases and Other Proteases

We first analyzed the granule proteases, as they are, in general, highly and uniquely expressed by MCs. This is why they are good markers of MC identity and phenotype. Three serine proteases and a carboxypeptidase constitute the major granule proteins of human MCs. Their transcript levels were very high and largely stable during a period of three weeks of in vitro culture, with the exception of chymase, CMA1, which was reduced by approximately 50% (Table 4).

The most highly expressed protease was cathepsin G (CATG), a neutrophil and mast cell-expressed protease with approximately 10,000 reads, followed by beta-tryptase (TPSB2) with 7000 reads, carboxypeptidase A3 (CPA3) with 4000 reads, and chymase (CMA1) with 2000 reads (Table 4). The expression of most other hematopoietic serine proteases was very low or absent in these cells, except for the delta- and gamma-tryptases (TPSD1 and TPSG1) and carboxypeptidase M (CPM), all with an average of 220 reads or lower (Table 4). A minor increase in cathepsin C (CTSC) from approximately 30 to approximately 75 reads was observed after culture (Table 4). This enzyme is involved in N-terminal cleavage and activation of the majority of granule proteases [42]. An approximately 10-fold upregulation, from an average of 20 to 200 reads, was also observed for another endoplasmic reticulum or Golgi-located protease, cathepsin W (CTSW), which is expressed by NK cells and cytotoxic T cells (Table 4) [43]. We did not detect expression of any of the T-cell granzymes, except for a low level of granzyme B (GZMB) in two of the in vitro cultures, with 2 and 11 reads, respectively (Table 4). No transcripts were detected for any of the neutrophil proteases (except cathepsin G as described above), including N-elastase, proteinase-3, or neutrophil proteases 4 (NSP-4), and there was no expression of myeloperoxidase or lactoferrin, which, except for NSP-4, are all found in relatively high amounts in mature neutrophils. Likewise, we did not find the expression of any of the classical eosinophil markers, ECP, EDN, and EPO.

Only very minor differences in expression between freshly isolated and cultured MCs were observed for the lysosomal proteases cathepsin B, D, and L1 (CTSB, CTSD, and CTSL1) (Table 4). In contrast, a marked upregulation was observed for two tissue metalloproteases, ADAMTS7 and ADAM14, from very low levels with a few reads in fresh cells to 100–150 reads in cultured cells (Table 4).

### 3.3. Transcript Levels for Protease Inhibitors

Protease inhibitors are important to control the effects of the massive amounts of proteases stored in MCs and other immune cells. We could observe some changes in the pattern of protease inhibitors after the in vitro culture of skin MCs. While no major change was seen for TIMP1, TIMP3, and cystatin 3 (CST3), a steep upregulation of cystatin 7 (CST7) and serpin B1 (SERPINB1) and a major downregulation of serpins E1 (SERPINE1) and H1 (SERPINH1) were observed (Table 5).

### 3.4. Transcript Levels for Fc Receptors and Other Cell Surface Receptors

All three chains making up the high-affinity IgE receptor FcεRI (i.e., FCERIA, MS4A2, and FCERIG) were upregulated two- to five-fold in culture. In contrast, a relatively strong, 10-fold, downregulation of FCGR2A was observed (Table 6). Interestingly, FCGR2A was the only Fc receptor, in addition to the high-affinity IgE receptor, expressed by these skin MCs. All other Fc receptors were essentially negative, including FCER2, FCGR1A, FCGR2B, FCGR2C, FCGR3A, and FCGR3B (Table 6). This is of major importance in view of the numerous reports in the literature of Fc receptor expression in MCs. We can show here that these human MCs essentially only express two Fc receptors, the high-affinity IgE receptor and one of the low-affinity receptors for IgG, FCGR2A.

There was a marked upregulation of the inhibitory receptor CD200R1 and a minor upregulation of the inhibitory receptors CD300A and MILR1, also called Allergin 1; however, in all three cases, the upregulation was from relatively low starting levels to levels between 100 and 200 reads (Table 6).

### 3.5. Transcript Levels for MRGPRX2, Purinergic, Cannabinoid, and Anti-Mullerian Hormone Receptors

For the majority of these receptors, including MRGPRX2, the ATP receptor P2RX1, the endothelin receptor B (EDNRB), and several other receptors, we could not detect any major changes in transcript levels upon in vitro culture. They showed a very stable expression with only small variations as shown in Table 7. We observed a minor increase in the cannabinoid receptor CNR2 and in the ATP/ADP receptor P2RX6. However, for the ATP/ADP receptor P2RY1, we could see a marked upregulation, although from very low starting values (Table 7).

### 3.6. Transcript Levels for Growth Factor receptors

No major changes in transcript levels were observed for the majority of receptors for cytokines and other growth and differentiation factors, including the stem cell factor receptor (KIT), the erythropoietin receptor (EPOR), the colony stimulating factor receptors CSF2RA and CSF2RB, the IL-33 receptor (IL1RL1), and others. However, for a few receptors including the IL-9 receptor (IL9R), we observed an upregulation from an average of 5 reads to 100–150 reads (Table 8). A marked downregulation of the lymphotoxin B receptor (LTBR), and several members of the TNF receptor family, including TNFRSF9 and TNFRSF21, was also observed (Table 8). Interestingly, relatively low levels of expression were seen for the majority of these receptors, rarely over 100–200 reads, except for KIT that was in the range of 1300–1400 reads in cultured MCs (Table 8).

The expression level of the TSLP receptor, CRLF2, was relatively high in ex vivo MCs and lower in cultured MCs, in accordance with previous results showing a difference in responsiveness to TSLP between these MC subsets (Table 8) [44].

### 3.7. Transcript Levels for MHC Class I and Class II Genes

No major change in expression was seen for the MHC class I genes, HLA-A, HLA-B, and HLA-C (Table 9). However, for the MHC class II genes, we detected a marked downregulation upon in vitro culture. The levels of MHC class II in freshly isolated cells were, however, very low compared to monocytes, as discussed previously [5]. Monocytes have been analyzed previously and their MHC class II expression levels were in the range of several thousand reads [45]. In the present study, the most highly expressed class II gene was HLA-DRB1, which in one MC preparation reached 146 reads (Table 9). Upon in vitro culture, the expression of the different class II genes was reduced to almost undetectable levels (Table 9). The transcription factor regulating class II expression, the CIITA, was also very lowly expressed and was reduced down to almost zero transcripts after in vitro culture (Table 9).

### 3.8. Transcript Levels for Enzymes Involved in Proteoglycan, Histamine, Prostaglandin, and Leukotriene Synthesis

No major change in the expression of the enzymes involved in proteoglycan synthesis was observed (Appendix A). In contrast, several enzymes involved in histamine prostaglandin and leukotriene synthesis were upregulated upon culture, including a minor (2-fold) upregulation of the histidine decarboxylase (HDC) from approximately 800 to 2000 reads, a 3–4-fold upregulation of the prostaglandin synthesis enzyme HPGDS from 600 to 2500 reads, and an almost 10-fold upregulation of the leukotriene synthesis enzyme leukotriene C4 synthase, LTC4S, from approximately 100 to 1000 reads (Table 10). No change in the expression of histamine receptor 4 (HRH4) expression was seen upon culture (Table 10). Interestingly, the only enzyme that showed a major downregulation was the phospholipase A2 -G2A (PLA2G2A) that is involved in the release of arachidonic acid from membrane phospholipids, generating the precursor of both prostaglandins and leukotrienes. However, PLA2G2A is a secreted form of phospholipase A2 and is probably of minor importance for the intracellular levels of arachidonic acid, and its role in MC biology is less well known. High levels of annexin 1 (ANXA1) mRNA were detected in these cells, with approximately 5000 reads in freshly isolated cells and a minor reduction to around 2000 reads in cultured cells (Table 10). Annexin 1 is a phospholipid binding protein also named lipocortin I, which acts as a negative regulator of phospholipase 2 and can thereby inhibit eicosanoid production and suppress inflammation [46]. A minor upregulation from approximately 700 reads to 1800 reads was seen for long-chain-fatty-acid-CoA ligase 4 (ACSL4), an enzyme with a role in lipid metabolism with preference for arachidonic acid [47]. A minor upregulation was also seen for the prostaglandin E receptor 3 (PTGER3). It is a receptor with a preference for prostaglandin E2, which typically increases FcεRI-dependent allergic responses in contrast to PTGER2 and PTGER4 [48]. We found no or very minor changes upon culture in the expression of PTGER2 and PTGER4 (Table 10). A low level of the complement component C2 and a marked upregulation of one of the anaphylatoxin receptors, C3AR1, was observed upon in vitro culture (Table 10).

### 3.9. Transcript Levels for Cell Adhesion Molecules

Several minor and a few major changes in the expression of cell adhesion molecules were observed, while the majority were relatively stable. The most pronounced upregulation was seen for integrin alpha 2B (ITGA2B), which went from a few reads to around 500 reads. A marked upregulation was also observed for CECAM 1 from around 10 reads to close to 300 reads (Table 11). A decrease in expression was instead seen for integrins alpha 5 and 9 (ITGA5 and ITGA9) which went from approximately 150–250 to around 25 reads (Table 11). An almost 10-fold decrease was also observed for L1CAM going from around 250 reads to between 20 and 50 reads after three weeks in culture (Table 11). L1CAM has important functions in the development of the nervous system, where it engages in cell–matrix and cell–cell interactions, e.g., by binding to integrins such as alpha Vβ3; its potential significance in MCs has been recently discussed [49]. In mouse peritoneal MCs, the most highly expressed cell adhesion molecules were instead Itgb1 with 444 reads, Itga4 with 404 reads, Itgb2 with 388 reads, Itga2b with 266 reads, and Itga9 with 217 reads [1].

### 3.10. Transcript Levels for Transcription Factors

The MC-related transcription factors GATA1, GATA2, MITF, and HEY1 showed no or very modest changes in expression upon culture, in agreement with most other transcription factors analyzed (Table 12). Also, no changes in transcript levels were seen for two more general transcription factors that have been shown to be of major importance for skin MC maintenance and for regulating GATA2, the evolutionary ancient CREB1 [39] and STAT5 [50,51], respectively (Table 12). However, we found examples of quite dramatic changes, and that was primarily in a few zinc finger and leucine zipper genes including KLF2, KLF4, TSC22D3, and ZFP36. These genes were highly expressed in the freshly isolated cells and strongly downregulated in culture (Table 12). KLF2 has a major role in erythroid and lung development [52]. However, its expression is most pronounced in lymphocytes [53]. KLF4 is highly expressed in non-dividing cells and downregulated upon the induction of cell division, thereby matching our data. The over-expression of KLF4 induces growth arrest [54]. ZFP36 is involved in mRNA stability and binds to AU-rich elements at the 3’ ends of mRNAs to increase their degradation, primarily of cytokine transcripts [55]. Its expression is very high in non-dividing myelocytes like granulocytes, monocytes, as well as MCs [53]. TSC22D3 may have a role in regulating the anti-inflammatory and immunosuppressive effects of IL-10 and corticosteroids [56]. Like KLF2, its expression is strongly enriched in lymphocytes [53]. Except for KLF4, the reason for the dramatic downregulation of these zinc finger and leucine zipper genes during transition from resting to actively dividing cells is not known, but this finding is consistent with what was previously reported for breast skin MCs [5].

### 3.11. Transcript Levels for Growth-Related Genes

Most of the growth-related genes were highly upregulated in culture, as indicated in the previous section where the growth-arrest-inducing zinc finger gene KLF4 was highly expressed in freshly isolated cells but almost totally absent from cells after in vitro culture (Table 12 and Table 13). Cell growth is apparently induced in culture and genes associated with the cell cycle are consequently increased in expression from low levels of a few reads to several hundred reads (Table 13). We observed potent upregulation of the histone H3 needed for new DNA assembly, cell cycle-regulated ribonucleotide reductase (RRM2), topoisomerase 2A (TOP2A), cyclin-dependent kinase (CDK1), a key player in cell cycle regulation, and of MKI67, a well-known marker for cell proliferation and ribosomal RNA synthesis (Ki-67) (Table 13) [57].

### 3.12. Transcript Levels for Cytokine and Growth Factor Genes

Most of the cytokines, chemokines, and other growth factor-related genes showed minor changes in expression. However, major changes were detected in a few entities. VEGFA, PDGFA, and IL-13 all showed a marked downregulation in expression upon in vitro culture, and this also applied to some degree to CSF1 (M-CSF) (Table 14). Both VEGFA and PDGFA dropped in expression by almost 100-fold, VEGFA from approximately 2000 reads to 20 reads, and PDGFA from around 350 to 7 reads (Table 14). There were also examples of the opposite, such as GM-CSF (CSF2), which showed a 10–100-fold increase to levels in the range of 300–800 reads, and IL-7, which increased 10 times but from very low initial levels (Table 14). A low level of transcription from the periostin gene (PSTN) was observed, a gene that has been associated with TH2 high asthma [58]. Slightly higher levels were seen for growth and differentiating factor 15 (GDF15) that was first identified as macrophage inhibitory cytokine-1, MIC-1 [59]. EMR2 is encoded by the ADGRE2 gene and a member of the adhesion GPCR family. This gene shows intermediate expression that does not seem to be substantially affected by culture (Table 10). A low level of the optineurin transcript (OPTN) as well as a high expression of TNFAIP3, also named A20, was noted. The latter encodes a zinc finger protein rapidly induced by tumor necrosis factor that seems to be importantly implicated in the downregulation of the inflammatory response induced by LPS [60] (Table 10). Its expression was strongly diminished upon in vitro culture.

### 3.13. Transcript Levels for Several Cluster of Differentiation (CD) Cell Surface-Expressed Proteins

Most of the CD molecules, including CD4, CD9, and CD14, showed relatively minor changes in expression upon in vitro expansion (Table 15). However, we found a few CD molecules showing more pronounced changes. These included CD63, which was increased by approximately 4-fold from around 300 reads to 1200 reads, CD274, also named PD-L1, which was decreased by approximately 10-fold from around 130 reads to 15 reads, and CD52, a 12 amino acid GPI-anchored peptide that may have an antiadhesive effect due to its high negative charge, which was increased by almost 40 times from around 20 reads to 800 reads (Table 15).

### 3.14. Transcript Levels for Circadian Clock-Related Genes

Most of the circadian clock genes show minor changes in expression, however, with some interesting exceptions. PER1 dropped in expression quite dramatically from almost 2000 to around 20 reads and TIMELESS increased by almost 50 times from very low levels around 2 reads to almost 100 reads (Table 16).

### 3.15. Transcript Levels for Other Proteins of Potential Interest

The expression levels of an extended list of genes of potential interest for the function of MCs can be found in Appendix A. In these tables, we list enzymes involved in proteoglycan synthesis, solute carriers, lipid transporters, calcium channels, sodium channels, potassium channels, other cell surface proteins, Siglecs, olfactory receptors, other receptors, cell signaling components, other enzymes, proteins related to vesicle transport, cytoskeletal and nuclear proteins, extracellular matrix proteins, oncogenes, and proteins of unknown function (Appendix A).

### 3.16. Transcript Levels for Heat Shock Proteins (HSPs) and Immediate Early Genes (IEGs)

A number of heat shock proteins and a few immediate early response genes including FOS, FOSB, and JUNB were highly expressed in the freshly isolated cells, some being the most highly expressed genes in these cells. These genes included the heat shock proteins HSPA1A, HSPA1B, HISPA6, and HSP90AB1, where HSPA1A reached over 75,000 reads, and FOSB, which had over 14,000 reads in one sample but was low in cultured cells (Table 17).

To understand whether the high expression of HSPs and IEGs in ex vivo MCs was an inherent property of the cells (that was lost in the conditions of the culture) or whether it was a stress response of the cells due to the isolation process, we performed two types of experiments.

Firstly, we made use of the fact that the stimulation of cultured skin MCs, e.g., by SCF or FcεRI aggregation, can induce a potent but transient IEG expression [10,39,40]. Here, we explored the kinetics of IEG and HSP expression following FcεRI aggregation. Indeed, expression of the two IEGs, FOS and FOSB, was induced and peaked after 30 min, but it diminished already after 90 min and was down to baseline after 24 h (Figure 2A). While IEGs are well known to be regulated by various stimuli, requiring CREB activity as an intermediary in skin MCs [39], it was unknown whether this also applies to HSPs. As displayed in Figure 2A, there was only marginal upregulation of HSPA1A and HSPA1B at 90 min (by ≈3-fold). Therefore, HSPs showed a distinct pattern of regulation vis-à-vis IEGs.

We then examined whether freshly extracted skin MCs rapidly downregulated IEGs and HSPs, which could suggest that the expression was primarily resulting from (unspecific) activation during isolation. The assumption was that if expression was the result of the isolation procedure (rather than a feature of the physiologic MCs), then expression would be at the level of cultured MCs after 1 day at the latest. However, this was not the case. As displayed in Figure 2B, the expression of HSP and IEG was still between ≈25 and ≈80-fold higher after 1 day versus in 3-week-cultured MCs. A significantly higher IEG expression was also detected after 2 days. The time-course, showing a drop in gene expression after 2 days versus 1 day, suggested that a subpopulation was gradually lost in the culture. In fact, 25–30% of skin MCs were lost during the first day, suggesting that their survival requires the skin microenvironment. Alternatively, MCs may actively downregulate expression of these genes in the new micromilieu.

In another series of experiments, cultured MCs were exposed to the conditions applied during isolation (i.e., digestive enzymes with or without shaking in a water bath). Compellingly, there was likewise no efficient induction of either IEGs or HSPs by incubation with collagenase/hyaluronidase/DNAse with or without shaking, establishing that the conditions MCs endure during isolation do not bring about the high expression found in ex vivo MCs (Figure 3). FOSB was a slight exception, but even after treatment with enzymes, its transcript abundance was at only 0.1–0.2% of the freshly extracted MCs, not explaining the high abundance in the latter.

## 4. Discussion

A detailed analysis of the total transcriptome of freshly isolated and cultured human skin MCs was performed to validate the quality of the in vitro-expanded MCs concerning their potential use as models of human MC biology. The cultured cells were found to be remarkably stable in their transcriptome compared to what has been observed for mouse BMMCs (Table 1) [26]. Mouse BMMCs represent a very immature version of mouse MCs where the expression levels of several important MC markers are very low compared to freshly isolated mouse peritoneal MCs, including the granule-stored chymase, Mcpt4 (mMCP-4), the tryptase Mcpt6 (mMCP-6), and the drug/neuropeptide receptor Mrgprb2 (Table 1) [26]. One additional important difference between these two model systems for studies of MC biology is that they represent cells of very different developmental origin. The mouse BMMCs originate from adult bone marrow, whereas the majority of human skin MCs have their origin primarily in an early wave of cells from the yolk sac [27,28]. A recent study of MCs from different mouse organs has shown that, also in the mouse, the organ of origin is a decisive factor for the resulting MC phenotype even though cells from different sources were exposed to identical microenvironments during their recent history [61].

The skin MCs used herein are human, which makes them better models of human biology than mouse MCs of any origin. The presented data harmonize with a previous analysis of skin MCs using another technology for transcriptome analysis, namely, deep-CAGE sequencing [5]. We have expanded this analysis here to include additional samples and another sequencing technology to verify the findings, and, most importantly, to extend to foreskin MCs, which had not been analyzed previously. Concerning granzymes, we did not detect expression of any of the granzymes under steady state conditions. However, some of them may be induced by LPS or IgE receptor crosslinking, as has been observed for GZMB, which was found to be highly inducible following FcεRI aggregation in human skin MCs in the Motakis study [5]. A fivefold increase in GZMB levels was also seen in mouse BMMCs upon LPS stimulation [1]. Also, mouse GZMC was actually upregulated from 0.1 reads to 100 reads, which corresponds to a 1000-fold increase but from very low starting levels [1]. However, during normal steady state conditions, granzymes seem to be almost totally absent from both mouse and human MCs.

In this study, the multiagonist receptor MRGPRX2 remained largely stable, although a slightly lower expression was detected in cultured MCs. This was in contrast to previous studies, demonstrating a prominent downregulation of this receptor at the transcript, protein, and functional levels in cultured skin MCs [5,62]. Even so, skin MCs still retain decent levels of MRGPRX2 in culture and can be used for MRGPRX2-related research, as substantiated by the findings herein. This is in marked contrast to mouse BMMCs that almost totally lack expression of the corresponding mouse receptor Mrgprb2 (Table 1). Expression of the dopamine receptor DRD2 was also observed in this dataset at similar expression levels as in previous studies [49].

Skin is a heterogeneous organ with different micromilieus prevailing in distinct locations. In the skin, MCs interact chiefly with endothelial cells of arterioles and fibroblasts, which shape the MC phenotype [63,64]. MCs interact with fibroblasts by different receptors, including but not limited to KIT, different integrins, and still unidentified receptor–receptor pairs [65,66,67]. The known receptors were also found to be expressed in our current study. Here, we analyzed the transcriptome in MCs derived from both foreskin and breast skin. We previously compared these two skin MC subsets by low-throughput techniques and found that the differences were rather small [68]. For example, breast skin MCs displayed slightly higher tryptase activity but lower histamine and chymase activity; however, there were huge variations within groups [68]. The very minor differences observed here, in global gene expression, between female and male samples further confirmed this concept. These findings suggest that despite their different sources (female, adult versus male, juvenile), major characteristics of skin MCs are surprisingly stable and are not strongly influenced by age, sex, or the precise location.

Concerning the cultured human skin MCs, one additional factor is the total number of cells. Upon in vitro culture, the cells start to proliferate, and we found that the expression of genes related to cell growth were increased (Table 13). After three weeks of culturing, the number of cells had increased by 6–10 times, resulting in more cells available for experiments. However, it is advantageous to use ex vivo skin MCs at least in a few experiments, to verify findings obtained with the precultured counterparts. The notion that expanded cells will resemble their respective prototype cell in the original tissue, even after culture, was recently reported also for mouse fetal skin or fetal liver-derived MCs [61]. However, our findings demonstrate a number of slightly unusual features that should be taken into consideration when using the in vitro-expanded human skin MCs in studies of MC biology (Figure 4 and Figure 5). Notably, the expression of the three genes for the high-affinity IgE receptor was increased by 2–5 times and there was an almost 10-fold reduction in the transcript level for the low-affinity IgG receptor FCGR2A. Moreover, there was an almost 10-fold upregulation of the leukotriene synthesis enzyme, leukotriene C4 synthase (LTC4S) (Figure 5). The demonstrated upregulation of high-affinity IgE receptor subunits and the leukotriene synthesis enzymes harmonizes with previous studies, and it is most likely explained by an induced response to the added SCF and IL-4 in the culture medium [18,38,62,69]. This also expands to IL-5 [70], GATA3 [71,72], and FCER1A [73]. There were also changes in cell adhesion molecules with a shift from integrin alpha 5 and 9 to beta 2A (Table 11). We found no or very minor changes in the enzymes involved in proteoglycan synthesis but a clear upregulation of some enzymes involved in leukotriene and prostaglandin synthesis (Table 10). In spite of these changes, the in vitro-expanded human skin MCs seem to be a very interesting model for studies of human MC biology. We can also say from this study that there are no or very minor differences between cells isolated from men or females. The only major sex-related differences we found were in the non-coding transcript XIST for X chromosome inactivation, which was found essentially only in the two female samples, and in the male-specific gene RPS4Y1, which is expressed from the Y chromosome (Table 3). XIST acts to cover one copy of the X chromosome, rendering it inactive [74].

For future studies including the design and evaluation of novel treatment strategies based on targeting MCs, there are some findings that need to be taken into serious consideration. One example is the very limited complexity in Fc receptor expression in human skin MCs. Attempts are made to use antibodies against the inhibitory IgG receptor FCGR2B for downregulating MC activation through the high-affinity IgE receptor. The possibility of inhibiting MC activation by FCGR2B crosslinking is probably very limited in view of the almost total absence of the transcripts for this receptor both in freshly isolated and cultured skin MCs (Table 6). However, this does not exclude that FCGR2B can be induced in skin MCs under disease conditions or that FCGR2B is expressed by human MCs in other tissues. Moreover, we did not detect the expression of any of the granzymes, which is in contrast to published work. Here, we found that the levels of all five granzymes in the freshly isolated skin MCs were very low, close to zero, indicating that granzymes have no or only a very limited role in human MC biology unless cells are stimulated.

The high expression of selective IEGs and HSPs was a hallmark of ex vivo MCs, while this quality was lost in culture, as reported previously [53]. Because the high expression of IEGs was likewise detected in monocytes, basophils, neutrophils, and other cells [53], we hypothesized that this was an inherent property of fully mature leukocytes, also encompassing ex vivo MCs, to be gradually lost outside of the body. Alternatively, it could have resulted from the isolation process.

Using several strategies, we found that the first hypothesis was most likely correct accounting for most of the effect. Firstly, we showed that mimicking the steps of isolation did not lead to a general upregulation of HSPs and IEGs in cultured MCs (only to a 20-fold increase in one gene), while the expression in cultured versus ex vivo cells was >10,000-fold lower (by RT-qPCR; up to ≈1000-fold by sequencing). This is in accordance with findings in the mouse, where the exposure of peritoneal MCs to digestive enzymes led to a maximally fivefold difference in a limited number of genes [75]. Interestingly, several IEGs were found among the regulated genes in this mouse study and FOSB was third in their list, in line with what we found here for FOSB (but not for FOS). However, this moderate increase can by no means explain the differences between ex vivo and cultured MCs, but it does indicate that a slight modulation during purification can occur in a gene-dependent manner. This upregulation is most likely transient, though. In fact, FOSB was down more strongly after 1 and 2 d compared to FOS (the latter not enhanced by digestive enzymes/shaking).

Additional evidence for an inherently high expression of HSPs and IEGs in skin MCs came from the fact that expression following MC purification did not diminish as rapidly as expression induced by MC stimulation. Indeed, the levels in ex vivo MCs were still increased at 1 or even 2 d vis-à-vis cultured MCs. In contrast, when IEGs were induced by FcεRI aggregation, the levels were back to their initial values after 1 d or less.

Our data are in line with the comprehensive FANTOM5 atlas. For example, granulocytes and monocytes expressed very high levels of certain IEGs, including FOS and JUNB. Both of the HSPs uncovered as high in skin MCs were also expressed in brain regions and reproductive organs (which were not exposed to digestive enzymes). Moreover, only some HSPs were highly expressed in skin MCs, while others were not; and the same applied to IEGs. Therefore, these genes are not expressed across the board in ex vivo cells/tissues but in clearly defined, cell- and tissue-specific patterns. For example, monocytes expressed substantial FOS but only little FOSB, while FOSB reached similar levels as in skin MCs also in basophils, eosinophils, Langerhans cells, and CD8^+^ cells [53]. Expression of HSPA1A/B was pronounced in the newborn brain, adult reproductive tissues (breast and vagina), heart, aorta, and adipocytes (in addition to influenza-infected macrophages). It is not entirely clear why primary cells accumulate such high levels of these transcripts. However, abundant expression seems to extend to the protein level in MCs and monocytes, whereby FOS (transcript and protein) abundance in primary monocytes was particularly striking [25]. In that paper, we compared the expression in primary myelocytes with corresponding cell lines (THP-1 and HMC-1), detecting much higher levels in primary cells, while reduction was a hallmark of actively cycling cells [25]. This was confirmed herein and extended to HSPs in MCs (yet not monocytes), emphasizing that each ex vivo cell expresses a unique combination of IEGs/HSPs. Regarding FOS specifically, it is described as indispensable in monocyte differentiation, increasing during maturation [76]. We assume that FOS and other factors play similar roles in MCs in vivo where they maintain the fully differentiated state and long-term survival within the cutaneous environment. In this regard, certain IEGs, HSPs, and several TFs like KLF2 and KLF4 (all much higher in freshly extracted MCs) may all be part of the same program.

In summary, this study provides a detailed transcriptomic characterization of in vivo matured human skin MCs in comparison with their in vitro-expanded counterparts. The findings confirm and extend our understanding of the phenotype of skin MCs in their normal tissue environment. Such information is also essential for the design and evaluation of novel potential treatment strategies aimed at targeting MCs in vivo. Based on previous and current findings, in vitro-expanded human skin MCs seem to be one of the best alternatives presently available for in vitro studies of the general in vivo function of human skin MCs. However, it should be noted that the large amounts of human skin needed to obtain a sufficient number of freshly isolated MCs is still an obstacle for the wider use of this model. To fully see the potential of these cells as in vitro models of human MC biology, future studies of the activation of these cells, both freshly isolated and cultured, by IgE receptor crosslinking and by activation of the MRGPRX2 receptor are needed. Some of these experiments are currently underway in our laboratories.

## Figures and Tables

**Figure 1 cells-13-00098-f001:**
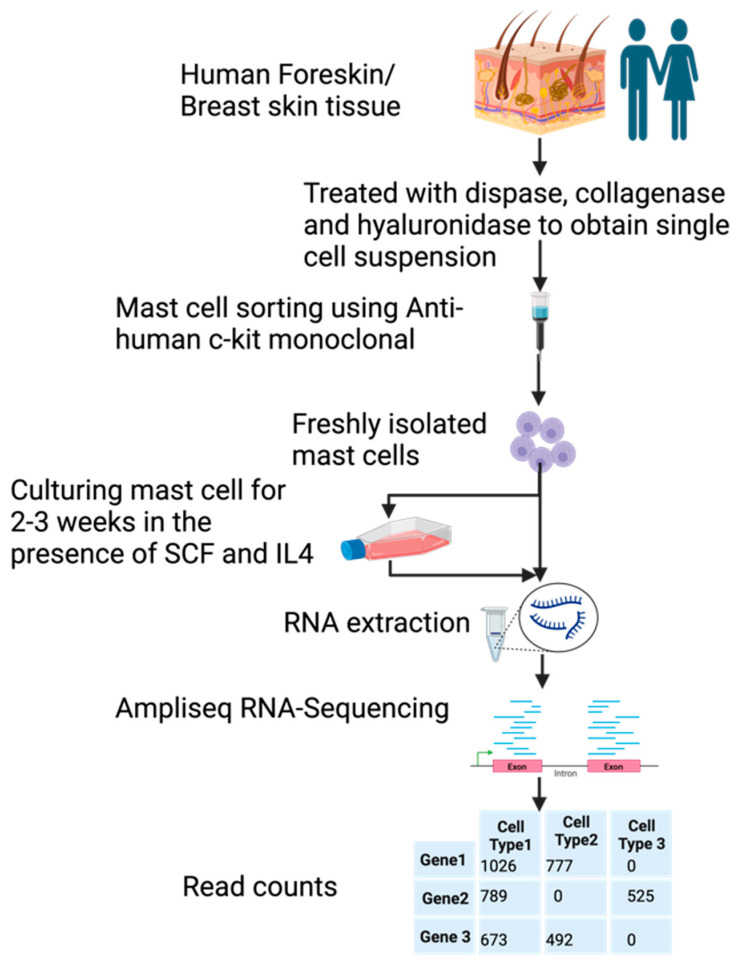
A schematic presentation of the experiment.

**Figure 2 cells-13-00098-f002:**
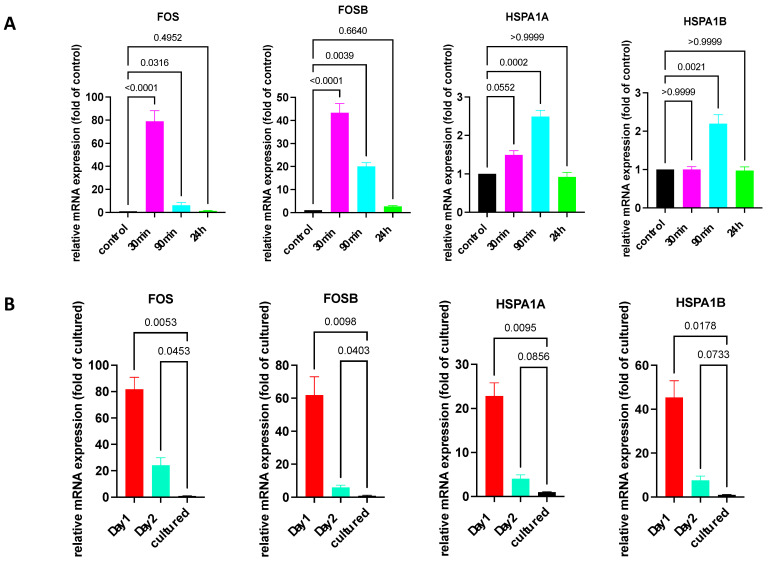
Expression of IEGs/HSPs quickly returns to baseline after FcεRI stimulation but not after purification of skin MCs. (**A**) Cultured skin-derived MCs were stimulated by FcεRI aggregation for the indicated times and expression of the respective genes was quantified by RT-qPCR and normalized to unstimulated control. N ≥ 7. (**B**) Skin MCs were kept in standard medium, 10%FCS, and no growth factors at 37 °C for 1 and 2 d following isolation, as described in methods. Cultured MCs were used after around 3 weeks in culture. Gene expression was studied as in (**A**). n ≥ 4. Normalized against 4 HKGs. The *p*-values are given directly in the figure.

**Figure 3 cells-13-00098-f003:**
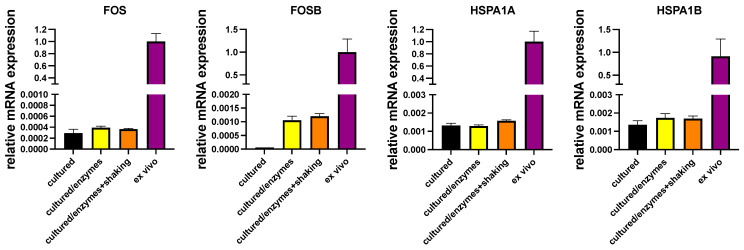
Conditions applied during isolation do not (efficiently) induce expression of IEGs/HSPs. Cultured skin MCs were exposed to digestive enzymes with/without additional shaking or kept in normal culture conditions (“cultured”). Expression of the respective genes in cultured skin MCs (exposed/non-exposed) and in freshly isolated skin MCs (ex vivo) was quantified by RT-qPCR and normalized to the control (black column). N = 2.

**Figure 4 cells-13-00098-f004:**
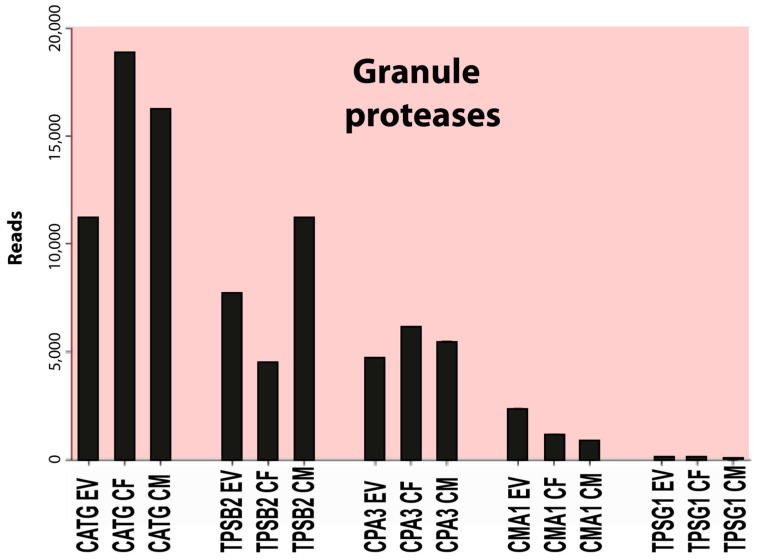
A summary of the major granule proteases of this transcriptional analysis of human skin MCs. EV stands for ex vivo or freshly isolated cells, CF for cultured female cells, and CM for cultured male cells.

**Figure 5 cells-13-00098-f005:**
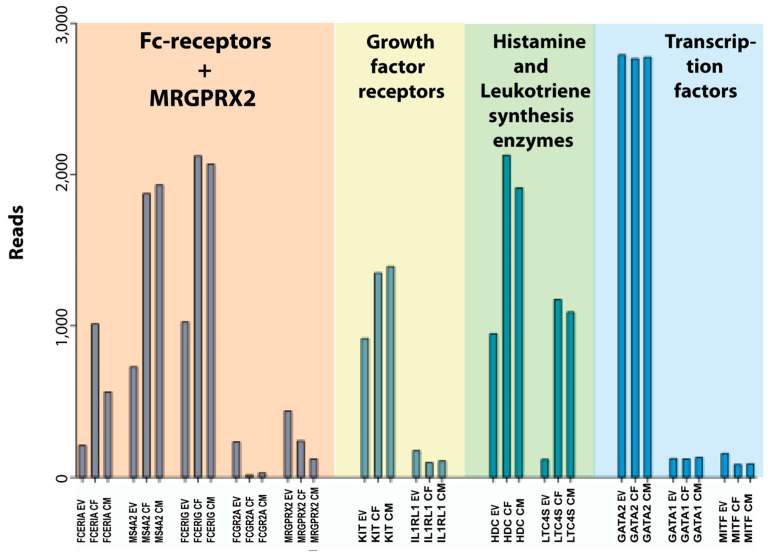
A summary of a few key genes of this transcriptional analysis of human skin MCs including Fc and MRGPRX2 receptors, growth factor receptors, histamine and leukotriene synthesis enzymes and transcription factors. EV stands for ex vivo or freshly isolated cells, CF for cultured female cells, and CM for cultured male cells.

**Table 1 cells-13-00098-t001:** Transcript levels in reads from an Ampliseq analysis of purified mouse peritoneal MCs and of a 3-week culture of mouse BMMCs.

Gene	Mouse Peritoneal MCs	Mouse BMMCs
**Proteases**		
*Cma1 (Mcpt5)*	45,221	15,683
*Mcpt4 (mMCP-4)*	31,290	7
*Tpsb2 (Mcpt6)*	67,773	3119
*Cpa3 (CPA-3)*	45,604	22,478
*Tpsab1 (Mcpt7)*	96	54
*Mcpt9 (mMCP-9)*	0	0
*CtsG (CTS-G)*	512	18
*Mcpt8 (mMCP-8)*	12	46
*Gzm B*	236	386
*Gzm C*	0.8	0.1
*Gzm D*	0.4	0
*Gzm E*	0	0.1
**Receptors**		
*FcεRI alpha*	252	1631
*Ms4a2 (IgE rec. beta)*	1297	4288
*c-kit*	720	833
*Mrgprb2*	899	8
**Transcription factors**		
*Gata1*	74	296
*Gata2*	2272	5205
*Gata3*	27	14
*Mitf*	370	144

**Table 2 cells-13-00098-t002:** Primer sequences for primers used in the quantitative PCR analysis described above.

Gene	Forward Primer (5′->3′)	Reverse Primer (5′->3′)
*FOS*	AGTGACCGTGGGAATGAAGT	GCTTCAACAGACTACGAG
*FOSB*	CTACGGAGCCTGCACTTTCA	AGCGAGTCCTCAAAGTACGC
*HSPA1A*	AGCTGGAGCAGGTGTGTAAC	CAGCAATCTTGGAAAGGCCC
*HSPA1B*	TGTAACCCCATCATCAGCGG	TCCCAACAGTCCACCTCAAAG
*ACTB*	CTGGAACGGTGAAGGTGACA	AAGGGACTTCCTGTAACAATGCA
*GAPDH*	ACATCGCTCAGACACCATG	TGTAGTTGAGGTCAATGAAGGG
*HPRT*	GCCTCCCATCTCCTTCATCA	CCTGGCGTCGTGATTAGTGA
*PPIB*	AAGATGTCCCTGTGCCCTAC	ATGGCAAGCATGTGGTGTTT

**Table 3 cells-13-00098-t003:** Transcript levels in reads from an Ampliseq analysis of freshly isolated MCs from foreskin and from purified MCs from breast skin and foreskin cultured for 2–3 weeks for two transcripts that are highly gender related.

	Freshly Isolated Cells			Cells Cultured for 2–3 Weeks			
	Foreskin (Male)			Foreskin (Male)		Breast Skin (Female)	
XIST	1	54	0	0	0	389	311
RPS4Y1	81	73	53	193	172	0.1	0.2

**Table 4 cells-13-00098-t004:** Transcript levels in reads from an Ampliseq analysis of freshly isolated MCs from foreskin and from purified MCs from breast skin and foreskin cultured for 2–3 weeks for the major granule proteases and a few additional lysosomal, tissue, and general proteases.

	Freshly Isolated Cells			Cells Cultured for 2–3 Weeks			
	Foreskin (Male)			Foreskin (Male)		Breast Skin (Female)	
Proteases							
CATG	9445	6194	12,156	18,847	13,470	16,285	21,210
TPSB2	7096	6177	7970	6791	15,604	3771	5143
CPA3	4403	3086	4976	6134	4812	5717	6465
CMA1	1713	1359	2958	954	655	1177	1160
TPSD1	418	67	145	78	62	196	156
TPSG1	86	49	107	39	53	73	88
CPM	140	81	19	48	52	87	91
GZMA	0	0	0.1	0.3	0.5	0.1	2
GZMK	0	0	0	0	0	0	0
GZMB	0.4	0.2	0	11	2	0.6	0.2
GZMH	0	0	0	0	0	0	0.2
GZMM	0	0	0	0	0	0	0
CTSC	23	28	36	74	83	83	72
CTSW	32	12	12	175	184	294	177
CTSD	2356	1723	2806	3513	4318	3354	5192
CTSB	241	315	334	499	608	495	517
CTSL1	48	158	55	29	27	22	28
PLAT (tPA)	136	288	137	317	158	164	218
PLAU (uPA)	102	38	100	18	12	16	20
ADAMTS7	80	58	56	41	45	68	53
ADAMTS14	2	1	3	98	159	108	141
ADAM12	9	6	4	144	144	158	146
PRSS12	38	59	98	10	3	3	4
CASP3	88	74	119	168	217	294	223
BACE2	103	49	39	399	425	473	434

**Table 5 cells-13-00098-t005:** Transcript levels in reads from an Ampliseq analysis of freshly isolated MCs from foreskin and from purified MCs from breast skin and foreskin cultured for 2–3 weeks for a number of protease inhibitors.

	Freshly Isolated Cells			Cells Cultured for 2–3 Weeks			
	Foreskin (Male)			Foreskin (Male)		Breast Skin (Female)	
Protease inhibitors							
TIMP1	407	357	502	692	630	569	982
TIMP3	1270	995	939	414	529	434	521
CST3	564	586	395	258	219	219	337
CST7	111	91	44	418	482	760	1210
LXN	27	132	49	16	17	16	13
SERPINB1	333	156	222	1652	1569	1669	1663
SERPINE1	200	339	115	9	0.7	6	2
SERPINH1	270	352	470	17	10	11	13

**Table 6 cells-13-00098-t006:** Transcript levels in reads from an Ampliseq analysis of freshly isolated MCs from foreskin and from purified MCs from breast skin and foreskin cultured for 2–3 weeks for Fc and other cell surface receptors.

	Freshly Isolated Cells			Cells Cultured for 2–3 Weeks			
	Foreskin (Male)			Foreskin (Male)		Breast Skin (Female)	
Fc receptors							
FCER1A (α)	214	127	207	494	633	987	1041
MS4A2 (β)	817	441	642	2129	1732	1966	1782
FCER1G (γ)	908	848	1140	2064	2074	1852	2391
FCER2	0	0	0	0	0	0	0
FCGR1A	0	0	0	0.3	0	0	0
FCGR2A	142	199	325	36	20	23	6
FCGR2B	2	0.4	0.2	0	0	0.1	0
FCGR2C	2	1	1	0	0	0.4	0
FCGR3A	0.3	0	0	0	0	0	0
FCGR3B	0	0	0	0	0	0	0
MILR1	57	29	68	162	145	135	204
CD200R1	24	6	5	84	85	109	88
CD300A	56	35	74	119	151	152	167

**Table 7 cells-13-00098-t007:** Transcript levels in reads from an Ampliseq analysis of freshly isolated MCs from foreskin and from purified MCs from breast skin and foreskin cultured for 2–3 weeks for MRGPRX2, purinergic, cannabinoid, and anti-Mullerian hormone receptors.

	Freshly Isolated Cells			Cells Cultured for 2–3 Weeks			
	Foreskin (Male)			Foreskin (Male)		Breast Skin (Female)	
Multiligand Pseudo-Allergy Receptor, Purinergic and Cannabinoid Receptors, Anti-Mullerian Hormone Receptors							
MRGPRX2	365	253	511	134	109	143	338
MAS1L	119	26	344	30	105	76	34
P2RX1	334	226	337	259	356	328	335
P2RX6	4	3	4	8	11	12	12
P2RY1	11	3	2	75	77	68	61
CNRIP1	113	74	81	241	224	382	124
DRD2	35	19	56	71	57	94	47
PAQR5	142	71	126	171	192	171	187
EDNRB	101	63	65	54	68	21	69
ADORA2B	8	6	5	17	30	21	23
ADORA3	16	3	5	14	20	10	12
CNR1	1	4	0.3	0	0	0.1	0
CNR2	0	0	0	10	11	13	8
AMHR2	59	26	57	86	104	85	101

**Table 8 cells-13-00098-t008:** Transcript levels in reads from an Ampliseq analysis of freshly isolated MCs from foreskin and from purified MCs from breast skin and foreskin cultured for 2–3 weeks for a number of growth factor, hormone, and retinoid receptors.

	Freshly Isolated Cells			Cells Cultured for 2–3 Weeks			
	Foreskin (Male)			Foreskin (Male)		Breast Skin (Female)	
Growth Factor Receptors, Hormone Receptors, and Retinoid Receptors							
KIT	1023	458	811	1381	1403	1347	1352
EPOR	105	38	112	58	63	69	73
CSF2RA (GM-CSFR)	14	6	6	3	0.8	1	1
CSF2RB (beta)	512	347	789	177	312	448	326
CRLF2 (TSLP-R)	194	427	257	87	76	85	77
IL1RL1 (IL33R)	125	289	226	115	99	96	99
BMPR1A	59	42	47	74	74	78	92
PTAFR	53	25	46	64	144	90	128
IL2RA	17	32	14	28	90	102	11
IL3RA	10	11	0.6	4	3	4	0.7
IL5RA	17	11	14	16	12	9	9
IL6R	69	98	27	7	13	16	15
IL6ST	124	252	115	61	62	65	110
IL9R	4	11	2	155	122	110	87
IL18R1	280	216	510	88	56	62	47
GPR34	39	9	14	206	224	332	304
MC1R	9	6	19	44	136	67	130
TNFRSF9	159	351	282	11	15	10	1
TNFRSF21	292	226	503	6	9	11	17
LTBR	74	65	43	8	10	3	4
ACVR1B	28	62	27	17	12	12	13
ADIPOR2	48	101	58	64	71	62	64
GABARAPL1	43	71	25	9	6	11	12
AGTRAP	115	50	109	201	253	215	293
ADRB2	561	590	832	262	252	244	278
RXRA	181	109	142	68	94	117	129
NRP1	58	71	26	150	149	171	213
NRP2	87	132	47	69	55	47	39

**Table 9 cells-13-00098-t009:** Transcript levels in reads from an Ampliseq analysis of freshly isolated MCs from foreskin and from purified MCs from breast skin and foreskin cultured for 2–3 weeks for MHC class I and class II genes.

	Freshly Isolated Cells			Cells Cultured for 2–3 Weeks			
	Foreskin (Male)			Foreskin (Male)		Breast Skin (Female)	
MHC and Related Genes							
HLA-A	344	326	253	167	201	231	329
HLA-B	297	188	70	93	106	338	48
HLA-C	552	241	200	251	196	449	475
HLA-DPA1	93	60	21	10	23	13	5
HLA-DPB1	39	28	9	1	0.3	0.2	0.4
HLA-DRA	75	65	19	0.3	0	0.4	0.2
HLA-DRB1	146	121	118	0.7	0.8	52	0
HLA-DQA1	17	8	4	0.4	0	0.2	0
HLA-DOA	2	0.6	0.4	0	0	0	0.1
HLA-DOB	0.1	0.1	0	0	0	0	0
CIITA	7	3	0.5	0.4	0	0.1	0

**Table 10 cells-13-00098-t010:** Transcript levels in reads from an Ampliseq analysis of freshly isolated MCs from foreskin and from purified MCs from breast skin and foreskin cultured for 2–3 weeks for enzymes involved in histamine, prostaglandin, and leukotriene synthesis.

	Freshly Isolated Cells			Cells Cultured for 2–3 Weeks			
	Foreskin (Male)			Foreskin (Male)		Breast Skin (Female)	
Histamine, Prostaglandin, and Leukotriene Synthesis and Receptors							
HDC	853	796	1044	1941	1880	1844	2409
HRH4	27	7	31	15	9	23	6
HPGD	1296	1021	1491	7877	8074	8857	9230
HPGDS	705	484	588	2133	2715	2728	2015
PTGS2 (Cox2)	523	392	353	881	364	293	248
PTGS1 (Cox1)	378	357	401	340	389	335	390
ALOX5	307	143	109	369	351	282	380
PLA2G2A	272	579	606	7	0	0	0
PLA2G3	0	0	0	0	0	0	0
PLA2G4A	18	12	9	33	43	20	28
LTC4S	129	60	105	923	1258	1111	1236
TBXAS1	26	18	26	104	129	128	108
ANXA1	3919	5423	6675	1938	1558	1907	1793
ACSL4	529	841	576	1961	1830	2027	1607
PTGER2	23	10	15	33	36	53	28
PTGER3	43	18	34	115	102	122	117
PTGER4	217	203	210	268	230	234	272
**Complement and Coagulation Components**							
C2	13	13	13	43	32	3	10
C3AR1	31	6	8	111	138	157	146
PROCR	36	19	14	20	22	12	18

**Table 11 cells-13-00098-t011:** Transcript levels in reads from an Ampliseq analysis of freshly isolated MCs from foreskin and from purified MCs from breast skin and foreskin cultured for 2–3 weeks for cell adhesion molecules.

	Freshly Isolated Cells			Cells Cultured for 2–3 Weeks			
	Foreskin (Male)			Foreskin (Male)		Breast Skin (Female)	
Cell Adhesion Molecules							
ITGA2B	4	3	6	547	479	649	654
ITGA3	98	75	92	34	23	18	34
ITGA5	162	358	176	25	21	25	22
ITGA9	148	127	156	23	17	28	22
ITGAV	29	148	24	25	13	20	18
ITGAX	86	231	76	83	99	81	115
PXN	180	127	147	31	29	34	37
SELPLG	60	21	34	135	228	262	209
ICAM1	146	53	15	38	42	52	37
L1CAM	304	232	233	52	58	24	20
FAT1	34	36	24	85	71	60	54
CEACAM1	11	6	14	261	309	301	254
NINJ1	181	266	123	19	24	37	16
PMP22	504	523	279	365	351	270	321
NTM	119	97	158	57	43	52	99
JPH4	93	62	127	80	109	72	94

**Table 12 cells-13-00098-t012:** Transcript levels in reads from an Ampliseq analysis of freshly isolated MCs from foreskin and from purified MCs from breast skin and foreskin cultured for 2–3 weeks for transcription factors.

	Freshly Isolated Cells			Cells Cultured for 2–3 Weeks			
	Foreskin (Male)			Foreskin (Male)		Breast Skin (Female)	
Transcription Factors							
GATA2	2859	1421	2719	2735	2815	2575	2955
GATA1	105	108	141	116	147	119	123
GATA3	1	2	1	29	12	10	15
MITF	195	64	118	94	83	80	87
HEY1	86	206	145	67	23	46	40
HES1	138	117	88	15	16	13	13
CREB1	75	83	54	67	64	73	71
STAT5A	80	56	70	78	86	86	65
STAT5B	111	100	156	127	128	150	133
BHLHE40	1449	969	715	976	1046	1612	1267
NFE2L3	18	19	27	20	21	22	18
PBX1	53	44	50	112	136	131	134
PHTF2	37	26	32	87	89	100	99
HOXB2	22	14	12	64	60	58	67
HOXB4	31	12	24	46	55	56	52
RUNX2	61	23	40	14	5	17	14
NR4A1	997	147	476	136	61	148	80
IKZF1	158	133	124	262	353	353	312
KLF2	2008	820	2018	8	8	3	10
KLF4	1751	1041	1184	0.7	0	0.6	0.1
TSC22D3	2856	2648	2859	5	8	7	6
ZFP36	15,395	10,607	11,923	122	53	289	138
ZNF618	54	26	74	61	69	67	73
ZNF521	43	19	35	70	56	69	77
ZCCHC24	53	65	46	60	77	70	77
ZMIZ1	190	300	224	135	158	153	159
GFI1	116	41	54	195	229	148	257
EGR3	494	194	902	422	220	379	248
MEIS2	275	193	320	287	260	272	285
STAT3	222	199	87	248	223	243	261
AFF2	95	48	90	91	90	89	104
TAL1	87	41	72	92	86	114	112
E2F8	2	1	1	39	32	31	20
FOXM1	0.3	0.3	0.2	111	81	75	55
GLI3	57	47	32	22	21	11	25
EPAS1	436	835	592	321	286	206	129
MAF	23	59	54	19	25	28	36
PTRF	465	707	280	175	125	198	140

**Table 13 cells-13-00098-t013:** Transcript levels in reads from an Ampliseq analysis of freshly isolated MCs from foreskin and from purified MCs from breast skin and foreskin cultured for 2–3 weeks for growth-related genes.

	Freshly Isolated Cells			Cells Cultured for 2–3 Weeks			
	Foreskin (Male)			Foreskin (Male)		Breast Skin (Female)	
Cell Growth-Related Transcripts							
HIST1H3G	8	9	10	1695	982	1145	645
HIST1H3J	5	4	4	481	319	363	174
HIST1H3F	4	2	1	229	238	277	185
RRM2	1	0.3	1	323	254	259	133
TOP2A	1	1	2	326	189	207	151
CDK1	1	1	1	140	120	108	74
MKI67	0.3	0.4	0.7	143	107	111	65

**Table 14 cells-13-00098-t014:** Transcript levels in reads from an Ampliseq analysis of freshly isolated MCs from foreskin and from purified MCs from breast skin and foreskin cultured for 2–3 weeks for cytokine and growth factor genes.

	Freshly Isolated Cells			Cells Cultured for 2–3 Weeks			
	Foreskin (Male)			Foreskin (Male)		Breast Skin (Female)	
Growth Factors							
VEGFA	1948	1323	2848	23	23	17	15
VEGFB	138	87	108	185	240	155	213
VEGFC	7	9	0.4	1	0.2	0	0.4
PDGFA	270	366	442	8	7	7	5
PDGFB	4	14	0.8	0	0	0	0
PDGFC	17	20	14	0.9	0.2	0.7	0.7
CSF1 (M-CSF)	1579	933	1736	466	232	561	274
CSF2 (GM-CSF)	9	61	6	276	360	627	799
CCL2	1518	1644	807	1789	925	1379	925
CCL4	34	117	47	29	34	1	98
CXCL16	263	160	310	117	156	185	252
LIF	659	307	1402	1798	1210	1582	1085
TGFA	31	21	41	58	59	65	87
TGFB1I1	137	149	75	27	13	8	11
TNF	138	124	213	19	43	5	8
TNFSF10	89	93	96	264	163	149	201
IL13	29	47	28	1	0.3	0.5	0.4
IL7	1	2	0.1	10	12	25	11
IL5	0.1	0	0	5	8	3	1
POSTN	19	24	1	40	4	32	25
GDF15	136	27	78	52	54	77	97
EMR2	154	207	168	127	109	148	118
OPTN	37	27	10	80	58	63	56
**Growth Factor-Induced Proteins**							
TNFAIP3	5148	1318	588	23	18	24	11

**Table 15 cells-13-00098-t015:** Transcript levels in reads from an Ampliseq analysis of freshly isolated MCs from foreskin and from purified MCs from breast skin and foreskin cultured for 2–3 weeks for several cluster of differentiation (CD) cell surface-expressed proteins.

	Freshly Isolated Cells			Cells Cultured for 2–3 Weeks			
	Foreskin (Male)			Foreskin (Male)		Breast Skin (Female)	
Surface Markers							
CD4	238	157	188	201	342	365	382
CD9	931	1078	997	914	833	896	770
CD14	22	26	16	21	27	9	36
CD22	146	110	118	411	585	730	738
CD33	42	28	70	140	100	199	179
CD34	4	2	3	0.1	0	0	0
CD52	29	20	9	954	776	833	697
CD63	265	306	270	1255	1092	1274	1264
CD68	355	305	275	1124	1240	686	999
CD274 (PD-L1)	106	158	130	17	17	15	10
CD276	17	32	28	32	44	34	41
CD109	16	13	7	31	32	30	31
PROS1	67	55	45	220	243	297	301

**Table 16 cells-13-00098-t016:** Transcript levels in reads from an Ampliseq analysis of freshly isolated MCs from foreskin and from purified MCs from breast skin and foreskin cultured for 2–3 weeks for circadian clock-related genes.

	Freshly Isolated Cells			Cells Cultured for 2–3 Weeks			
	Foreskin (Male)			Foreskin (Male)		Breast Skin (Female)	
Circadian Proteins							
PER1	2028	1197	2104	26	18	26	23
CLOCK	20	17	13	46	39	41	43
PER2	45	54	29	10	12	11	17
TIMELESS	4	2	1	97	71	98	62
PER3	39	21	15	40	36	25	39
ARNTL	5	3	1	13	15	15	14
NR1D1	45	36	25	13	13	10	11
NR1D2	37	33	12	31	31	34	32

**Table 17 cells-13-00098-t017:** Transcript levels in reads from an Ampliseq analysis of freshly isolated MCs from foreskin and from purified MCs from breast skin and foreskin cultured for 2–3 weeks for heat shock proteins and immediate early genes.

	Freshly Isolated Cells			Cells Cultured for 2–3 Weeks			
	Foreskin (Male)			Foreskin (Male)		Breast Skin (Female)	
Stress and Growth Response							
FOSB	13,651	8861	14,157	49	5	39	6
FOS	6395	3043	2026	140	51	158	60
JUN	30	17	16	0.3	0.2	0.2	0.1
JUNB	5676	3118	3982	231	250	331	315
HSPA1A	33,988	35,501	75,250	211	293	259	293
HSPA1B	11,715	5094	23,085	65	85	57	45
HSPA6	417	396	950	0	0	0.4	0
HSPH1	885	423	702	115	111	94	108
HSP90AB1	2350	2500	4641	391	430	432	479
PLAUR	888	415	1012	65	41	70	47
CLU	958	799	797	1523	1963	2460	2376
CREB3L2	338	307	323	375	471	401	350
IER3	1186	1693	836	1059	749	1175	1143
EGR3	494	194	902	422	220	379	248

## Data Availability

All information of importance for the conclusions are present in the manuscript.

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
