# Peer review of "Cultures of Human Skin Mast Cells, an Attractive In Vitro Model for Studies of Human Mast Cell Biology"

_cells, 2024, doi:10.3390/cells13010098_

Round 1

Reviewer 1 Report

Comments and Suggestions for Authors

The manuscript details the total transcriptome of freshly isolated human skin mast cells (MCs) and in vitro cultures of these cells. The study also compared previously reported mouse peritoneal and bone marrow-derived MCs (BMMCs) transcriptomes. Although data may be helpful for mast cell community researchers, it would have been an improvement to include transcriptomics after mast cell activation (IGE dependent, MRGPRX2) and compare fresh cells to culture cells. This data would have been helpful to assess the relevance of working with fresh versus cultured mast cells. A comment on those aspects should be made in the manuscript. The quality of the tables should be improved, and the columns of the different samples should be separated.

Author Response

Our aim is to continue this analysis of freshly isolated human mast cells and of in vitro cultured cells for their response to IgE receptor crosslinking and for activation through MRGPRX2. However, no such data is yet available. We have added a comment of this interesting future studies in the end of the discussion section (marked in red).

The lack of separating lines in the tables is a modification made by the production office of Cells. We have separating lines in the original manuscript but these were removed during the formatting of the PDF version of the manuscript. We will instruct the editorial office to insert them again.

Reviewer 2 Report

Comments and Suggestions for Authors

The paper is a highly comprehensive document on gene expression-total transcriptome- in mouse peritoneal mast cells as compared to BMMC, and of freshly isolated foreskin mast cells as compared to cultured breast skin and foreskin mast cells. Methods are up to date and the results are clearly presented and discussed.

Author Response

Thanks for kind comments. We have made some changes suggested by the other two reviewers primarily in the tables to insert separating lines for clarity, as these lines were present in the submitted manuscript but removed by the production team.

Reviewer 3 Report

Comments and Suggestions for Authors

This referee is thinking that this study provides valuable information for mast cell study, though there are some points to be revised.

1) Tables 4 – 17: This referee interpreted doners of “Cells cultured for 2-3 weeks (Foreskin [Male])” were not identical with those of “Freshly isolated cells (Foreskin [Male])”. To eliminate individual variation, “Cells cultured for 2-3 weeks (Foreskin [Male])” should be cultured using partial component of “Freshly isolated cells (Foreskin [Male])”.

2) Tables 4 – 17: It seems to be confusing which rows are corresponding to Freshly isolated cells / Cells cultured for 2-3 weeks or Foreskin (Male) / Breast skin (Female).

3) Table 1: This referee could not understand the significance of this table using “Mouse peritoneal MCs” and “Mouse BMMCs”. And, this referee is thinking that the data using “Freshly isolated mast cells from mouse neonatal skin” and “Cells cultured for 2-3 weeks from mouse neonatal skin” is suitable for this table.

4) Abstract: This referee thinking that the differences between “Cells cultured for 2-3 weeks” and “Freshly isolated cells” should be noted more clearly.

5) Line 194 – 195: This referee could not understand what “three separate donor pools, 12, 7, 6 for the three fresh and for the cultured 5- 15” is meaning.

END

Comments on the Quality of English Language

Minor editing of English language required.

Author Response

Separating lines between samples were present in the submitted manuscript but these were removed by the production office. I will tell them to reinsert them in all tables.

We have also modified the abstract to more clearly present the differences between freshly isolated and cultured mast cells (marked in red).  

Table 1 is where we have data using the same technology as for the human cells in this study PMCs and BMMCs are also the model systems that is the most commonly used in the field why we think it’s the most appropriate data to compare with. This explanation has been added to the introduction (marked in red).

We have also clarified the text concerning the numbers of donors for each experiment as relatively few cells can be obtained from one foreskin and also often from a single breast donor why several samples have to be combined to obtain sufficient number of cells (marked in red).

Reviewer 4 Report

Comments and Suggestions for Authors

Mast cells are important therapeutic targets in different immunopathological conditions. Mast cell lines are suitable as models only to a limited extent. Therefore, the search for cultured primary cells as suitable model systems continues.

I really like the authors' idea - to make a comparison between freshly isolated and cultured cells in terms of gene expression. Even if subtle differences between individual functional skin MC subpopulations cannot be detected in this way, the transcriptome data will help us to understand functional characteristics of skin mast cells better.

However, there are some shortcomings that need to be resolved. Here my suggestions to improve the manuscript:

1)      I would suggest structuring the gene expression data in the tables in different way - first the data of the freshly isolated cells, then the cultured cells of the male donors and then the female donors. Incidentally, the authors already use this sequence in the text (lines 193-197)

2)      It is unclear to me why all tables have the same title.

3)      Since the whole manuscript is based on observations made from 7 samples, I would ask the authors to provide more data on the samples. How many cells were isolated? Percentage of dead cells in the samples? Cell morphology? Content of histamine /proteases?

4)      The cells were cultivated in the presence of IL-4 and SCF. Are IL-4 induced genes more strongly represented in samples of cultured cells, or is the IL-4 gene signature already present in freshly isolated samples?

5)      Data generated in mouse show that in the skin the mast cells interact primarily with endothelial cells of the arterioles and skin fibroblasts (Kaltenbach et al 2023). Furthermore, fibroblasts are shaping skin mast cell response (Di Nardo 2023). Can the authors derive similar observations from the analyses of the transcriptome data of human skin mast cells? Is the integrin pattern observed by transcriptome analysis comparable between mouse and human mast cells?

6)      In my opinion, Table 1 with the data on the mouse cells is not admissible because it has already been published in another article.

7)      Expression data of some genes are shown in the tables, without further comments in the text. Would it be possible to comment on the expression analyses in more detail? (Genes of the complement system, periostin, GDF15, ADGRE2(EMR2), optineurin, A20 (TNFAIP3) are the examples).

8)      Rönnberg et al (2014), Pardo et al (2007), Strik et al (2007), Bladergroen et al (2005) showed that human and murine mast cells express granzyme B, granzyme B inhibitor or Granzyme H.

Please check the spelling

-          Line 119,

-          Lines 194-195,

-          Line 169.

Author Response

Due to added figures in the response letter, we include a PDF copy of the response letter.

Round 2

Reviewer 3 Report

Comments and Suggestions for Authors

None.